# A Deep Neural Network-Based Method for Early Detection of Osteoarthritis Using Statistical Data

**DOI:** 10.3390/ijerph16071281

**Published:** 2019-04-10

**Authors:** Jihye Lim, Jungyoon Kim, Songhee Cheon

**Affiliations:** 1Department of Healthcare Management, Youngsan University, Yangsan 626-790, Korea; limjiart@ysu.ac.kr; 2Department of Computer Science, Kent State University, Kent, OH 44242, USA; jykim2@kent.edu; 3Department of Physical Therapy, Youngsan University, Yangsan 626-790, Korea

**Keywords:** osteoarthritis, prediction, deep learning, feature extraction

## Abstract

A large number of people suffer from certain types of osteoarthritis, such as knee, hip, and spine osteoarthritis. A correct prediction of osteoarthritis is an essential step to effectively diagnose and prevent severe osteoarthritis. Osteoarthritis is commonly diagnosed by experts through manual inspection of patients’ medical images, which are usually collected in hospitals. Checking the occurrence of osteoarthritis is somewhat time-consuming for patients. In addition, the current studies are focused on automatically detecting osteoarthritis through image-based deep learning algorithms. This needs patients’ medical images, which requires patients to visit the hospital. However, medical utilization and health behavior information as statistical data are easier to collect and access than medical images. Using indirect statistical data without any medical images to predict the occurrence of diverse forms of OA can have significant impacts on pro-active and preventive medical care. In this study, we used a deep neural network for detecting the occurrence of osteoarthritis using patient’s statistical data of medical utilization and health behavior information. The study was based on 5749 subjects. Principal component analysis with quantile transformer scaling was employed to generate features from the patients’ simple background medical records and identify the occurrence of osteoarthritis. Our experiments showed that the proposed method using deep neural network with scaled PCA resulted in 76.8% of area under the curve (*AUC*) and minimized the effort to generate features. Hence, this methos can be a promising tool for patients and doctors to prescreen for possible osteoarthritis to reduce health costs and patients’ time in hospitals.

## 1. Introduction

Osteoarthritis (OA) is the most common joint disease that affects 10–15% of adults worldwide and causes chronic pain and disorders [1,2]. The World Health Organization estimated that 18.0% of women and 9.6% of men aged 60 years or older had symptomatic OA in 2015. Among people with symptomatic OA, 80% have some type of limitation in mobility, and 25% are unable to perform their daily activities [3]. According to the 2012 Korean National Health and Nutrition Examination Survey (KNHANES) results, OA prevalence in people aged 50 years and above was 3.3% in males, and 16.0% in females [4]. With the average life expectancy gradually increasing, the prevalence of osteoarthritis may also increase depending on the size of the aged population. OA is a disease that brings physical and psychological sequelae to individuals and, in addition to that, imposes a burden on the society and economy [5].

The primary symptoms of OA include joint pain, stiffness, and limitation of movement. According to previous studies, the major risk factors for developing osteoarthritis are obesity, metabolic diseases, age, gender, ethnicity, nutrition, smoking, bone density, and muscle function [5,6]. OA is a slowly progressing disease, and since damaged cartilage cannot be restored to its normal state, prevention should be the focus for treatment [7]. Thus, early examination and diagnosis are more important than anything for the health of individuals and communities. Symptomatic OA was one of the diseases with the most expensive treatment in U.S. hospitals in 2008, and the national cost of hospital treatment by arthroplasty surgery reached approximately $40 billion [1].

Recently, big data have spread socially, and many researches have been trying to identify multidimensional factors influencing health and disease by using machine learning algorithm [8,9]. In recent years, deep learning has revealed an exciting new trend in the field of machine learning. The theoretical foundations of deep learning are well-based on the classical neural network (NN) literature. However, differently from the more traditional use of NNs, deep learning explains the use of many hidden neurons and layers, typically more than two, as an architectural advantage combined with new training paradigms [10]. Computer algorithms will support optimal clinical decision-making. The deep learning concept lies at the basis of many learning algorithms: models (network) composed of many layers transform input data (e.g., variables) into outputs (e.g., disease present/absent), while learning increasingly higher-level features [11].

Deep learning has had the greatest impact within the field of health informatics. Health informatics focuses on the analysis of large, aggregated data, with the aim to enhance and develop clinical-decision support systems or assess health data, for both quality assurance and accessibility of health care services. Electronic health record datasets are extremely rich sources of patient information, which include medical history details such as diagnoses, diagnostic exams, medications and treatment plans, immunization records, allergies, radiology images, and laboratory results [9]. Efficient analysis of these big datasets would be able to provide valuable insights into disease management [12].

In many previous studies, there were limitations because of the use of only several variables related to osteoarthritis or the lack of big data use [13,14,15]. Therefore, research is needed to develop a predictive model of the prevalence of osteoarthritis, which is socially increasing the burden of disease, using the deep learning method based on big data. This study aimed to analyze associated factors of osteoarthritis and to develop a predictive model of osteoarthritis by using the 2015–2016 Korea National Health and Nutrition Examination Survey (KNHANES) data. The results from this study may provide significant data that could aid the development of regional and national health policy designs.

The present study intends to develop a deep learning model to detect the occurrence of OA based on the statistical data of medical utilization and health behavior information. Most of the previous methods have the advantage of being relatively accurate and specific to the type of diseases. On the other hand, they have some limitations, such as inefficiency, and being uncomfortable and time-consuming. More details about previous works are shown in Section 2.

## 2. Related Work

There have been several studies which have used deep learning methodologies based on diverse data to detect several problems [16,17,18,19,20]. These include studies on detection or prediction of OA with machine learning or deep learning, conducted with diverse approaches and datasets. Shaikhina et al. [21] proposed an NN model for osteoarthritic bone fracture risk stratification and a decision-tree model for the prediction of antibody-mediated kidney transplant rejection. Although they achieved high accuracy, they used small datasets (35 bone specimens and 80 kidney transplants). Kovanova et al. [22] investigated trabecular bone in OA using NN with 25 available samples. Antony et al. [23] applied Convolution Neural Network (CNN) to quantify the severity of knee OA. They pre-trained dataset on ImageNet and turned parameters on knee OA images. Kobashi et al. [24] investigated a post-operative knee function prediction model using a PCA-based method with 2-D and 3-D X-ray data (52 OA patients). Ashinsky et al. [25] evaluated the ability of a machine learning algorithm to classify magnetic resonance images (MRI) of 65 OA patients. Lazzarini et al. [26] proposed an analytic pipeline based on machine learning to predict the 30months incidence of knee OA using diverse data, including clinical variables, questionnaires, biochemical markers, and images. Xue et al. [27] examined the diagnostic value of CNN with 420 hip X-ray images to detect hip OA. Tiulpin et al. [28] also used CNN to detect knee OA with X-ray images from 3000 subjects. Du et al. [29] explored features in knee MRI for OA prediction with PCA with several machine learning algorithms, such as NN, support vector machine, and random forest. Hirvasniemi applied machine learning to predict the incidence of radiographic hip OA using 986 images. Brahim et al. [30] applied machine learning (multivariate linear regression) to detect early knee OA using knee X-ray. Most of the previous studies focused on image data with machine learning or deep learning with image data, such as MRI or X-ray. However, to the best of our knowledge, the studies of prediction using statistical data with machine learning or deep learning are rare.

## 3. Materials and Methods

### 3.1. Data Source

The data used in this study were taken from the 2015–2016 KNHANES, performed by Korea Centers for Disease Control and Prevention (KCDCP). KNHANES, based on the National Health Promotion Law, was conducted nationwide as a cross-sectional study. Households were randomly selected for participation and sampled using multi-stage stratification based on geographical areas [6]. The Institutional Review Board (IRB) of the KCDCP (2015-01-02-6c) approved the use of the 6th KNHANES raw data. The IRB of the KCDCP operated according to the guidelines of the Declaration of Helsinki (2000).

Of the 15,530 participants who included responses regarding medical utilization and health behavior in the National Health and Nutrition Survey in 2015–2016, 6629 were adults aged over 50. Among these, we excluded participants who did not provide important information. A total of 5749 individuals were finally included in this study. The dataset was split into two parts: training (66%) and testing (34%) data, as shown in Figure 1.

### 3.2. Main Variables

Patients with OA were identified using the following question: “Have you ever been diagnosed with osteoarthritis by a doctor?” The independent variables used demographic and personal characteristics, such as gender, age, region, marital status, education, and household income. The region variable was divided into Seoul, metropolitan, Gyeonggi, and other areas in South Korea. Household income levels were split into categories (highest, upper middle, lower middle, lower) based on the quartile of household equalization income. Lifestyle- and health status-related variables included smoking status, BMI, obesity, alcohol intake, self-reported status of health, physical activity, and medically diagnosed chronic disease. Chronic diseases included hypertension, diabetes mellitus, dyslipidemia, stroke, myocardial infarction (MI), angina pectoris, and osteoporosis, which were diagnosed by a medical doctor. Smoking status was classified according to three types of smokers, i.e., smoker, former smoker, and non-smoker. The obesity variable was divided into a low-weight group for body mass index (BMI) lower than 18.5, a normal-weight group for BMI 18.5–25.0, and an obese group for BMI higher than 25.0. The self-reported health status was classified as very good, good, moderate, poor, and very poor. Physical activity was defined as over 10 min of medium-strength vigorous physical activity, which involved breathing slightly fast or faster than usual.

### 3.3. Methods

In this study, we adopted and present a deep learning approach as a supervised learning model for predicting the presence of osteoarthritis in subjects aged 50 years and older using statistical data, such as gender, age, household income, marital status, smoking status, alcohol intake, BMI, physical activity, number of chronic diseases, etc. Deep Neural Network (DNN) and scaled Principal Component Analysis (PCA) were employed to automatically generate features from the data and identify risk factors of osteoarthritis prevalence. 

Figure 2 shows the proposed system architecture of the DNN with scaled-PCA: (1) KNHANES data, including 24 raw features divided into training (66%) and testing data (36%); (2) data preprocessing based on PCA and a scaler was applied to convert categorical variables into continuous variables and generate models for testing the data; (3) the DNN model was also trained on the basis of scaled-PCA variables and generated the trained DNN model; (4) the predicted results were evaluated with the ground-truth data labeled by clinicians. We divided the training and testing data; therefore, there were no overlapping patient data.

We used four terms, i.e., true positives (*TP*), true negatives (*TN*), false positives (*FP*), and false negatives (*FN*), to compute the performance parameters and evaluate the developed models. *TP* was the correctly predicted OA, and *TN* was the correctly predicted non-OA. *FP* and *FN* were the incorrectly predicted OA and non-OA, respectively. Traditionally, the performance of detection or prediction algorithms is measured by the Accuracy (*Acc*). However, the two-classed data set we used in this study was imbalanced. In the data sets, there were more non-osteoarthritis data (4509) than osteoarthritis data (1240). To address this class imbalance problem, we used three additional metrics for proper evaluation: sensitivity (*Sn*), specificity (*Sp*), and positive-predictive value (*PPV*). Sensitivity is a probability of the predicted cases of osteoarthritis status, specificity is a probability of the predicted cases of non-osteoarthritis status, and positive predictive value is a probability of the corrected prediction of the osteoarthritis status. *Sn*, *Sp*, *PPV*, and *Acc* were calculated by using the following equations:*Sn* = *TP*/(*TP* + *FN*)(1)
*Sp* = *TN*/(*TN* + *FP*)(2)
*PPV* = *TP*/(*TP* + *FP*)(3)
*Acc* = (*TP* + *TN*)/(*TP* + *FN* + *FP* + *TN*)(4)

### 3.4. Preprocessing

PCA is a standard method of modern data analysis and a simple and non-parametric methodology for extracting relevant information from confusing datasets. In general, PCA is a preprocessing tool to obtain new features in a compact way, compared with the original features for extracting reduced dimensions to find the hidden or simplified structures that enable fitting into classification algorithms [31]. However, the dataset used in this study consisted of major categorical/binary and minor continuous variables. The categorical/binary variables were not appropriate feature inputs for the DNN classifier, because of the lack of detailed information. We set the maximum attribute filter of PCA to convert all 24 binary or categorical variables into 24 continuous variables, in order to minimize discrete types of data. Figure 3 shows first and second principal components of four different PCAs, including (a) normal PCA, (b) PCA with standard scaler, (c) PCA with min/max scaler, and (d) PCA with quantile transformer scaler. According to our experiment, the combination of DNN and PCA with quantile transformer scaler showed the best performance.

### 3.5. Architecture of Deep Neural Network

We used simple feed-forward neural networks trained with the standard backpropagation algorithm for our deep (eight hidden layers) learning models. For each tested DNN, multiple hyper-parameters were adjusted, including the number of hidden layers, the number of neurons in each layer, the choice of the activation function, the choice of the optimization method, and regularization techniques. The best DNN in the ensemble had eight hidden layers with 90, 90, 90, 90, 90, 90, 90, and 20 neurons in each, respectively. The last layer, with one neuron, corresponded to the regression output. The optimization loss function was accuracy with regularization terms. The DNN used the ReLU activation function [32] in each layer and Dropout [33] with probability of 0.2 after each layer. They were trained with Adam optimization [34], because the Adam optimizer is more robust when using hyper-parameters and usually works very well empirically. To further cope with over-fitting and develop a more stable convergence of the models, we used the Batch normalization technique [35] after the first two layers.

All models were implemented using Keras [36] with the TensorFlow [37] backend. The binary cross entropy was adopted as the loss function for the task of classifying the occurrence of osteoarthritis. Since the classes of the dataset were imbalanced, a class weight scheme was used to fix this issue. Through the use of the class weight, the minority classes became more important. 

## 4. Result and Discussion

To perform this study, we obtained a dataset of 62,419 osteoarthritis records, where each record consisted of a person’s occurrence of osteoarthritis, and 24 features from KNNDB in South Korea. The resulting dataset was split into training and test sets, comprising 3794 and 1955 samples, respectively. In the training set, we set 30% of training samples as a validation samples. Among various experimental settings, the best DNN architecture was 8 hidden layers with 90 neurons, and the epoch size and batch size of the training setting were 500 and 100, respectively. Table 1 provides the confusion matrix results from the proposed DNN algorithms with the scaled PCA preprocessing. Thus, considering all the effective parameters of the developed model, the most suitable probability threshold of osteoarthritis was 0.35, with classification model accuracy of 71.97%, sensitivity = 66.67%, specificity = 73.35%, and positive predictive value = 39.53%. Compared to common performance metrics, such as sensitivity, specificity, positive predictive value, and accuracy, the AUC is a better way to judge an algorithm performance by using one single value [38]. The proposed methodology showed an *AUC* of 76.8%. The receiver operating characteristic (ROC) curve for the predictive performance of DNN with scaled-PCA is shown in Figure 4. 

The combination of the scaled-principal components and the trained DNN estimated the occurrence of osteoarthritis. We further examined the correlation coefficient between each detected feature regarding statistical features and osteoarthritis. However, it did not clearly show a detailed relationship between principal components and the occurrence of osteoarthritis. Table 2 shows the correlation coefficients (r-values) of 24 initial features, where the top two best correlation values (±0.25) are highlighted in bold, and the three other interesting results (±0.2) for each measure are highlighted in italic or red. Overall, on average, the features of sex and osteoporosis had a high correlation with the occurrence of osteoarthritis.

We considered tuning the hyper-parameters during the major process to implement the appropriate DNN for detecting OA, such as the number of nodes and depth of DNN. Although tuning the parameters is important, no general rules have been determined, thus we had to train many combinations, such as 4, 6, 8, and 10 layers, with 70, 80, 90, and 100 nodes by trial and error. Another important issue is overfitting. We adopted two methods, i.e., the Dropout and Batch normalization techniques. Batch normalization is used to prevent vanishing feedforward data like good weight initialization, and dropout is used to minimize the effects of certain hidden nodes with weights. 

The proposed method predicted the occurrence of OA with indirect or limited data, such as the statistical data of medical utilization and health behavior information. This can be an advantage for possible patients to prevent future medical costs and reduce the time for diagnosis. There are some limitations, such as the lack of input data separation and the absence of progressive data for OA. This study was based on a survey which had several defects. Most input features were of binary type. Although we applied scaled-PCA to improve data separation, additional feature inputs may be required. In addition, our results only apply to subjects with determined OA disease, since we excluded subjects who were receiving treatment for OA. This issue may reduce the overall accuracy of the prediction model.

## 5. Conclusions and Future Work

We propose a deep learning model with scaled PCA for automatic osteoarthritis prediction, based on medical utilization and health behavior data (of 5749 subjects) without any hand-crafted features, and validated it in a large population. The most important finding of our study is the early detection of patients at high risk of OA, who need additional checkup and appropriate treatment before OA exacerbates. The proposed method can also save a lot of time and effort in designing, computing, and selecting the features for the predictive analysis. Since the input data from the datasets were simple and had a low resolution, such as binary or limited number of categories, we employed DNN to learn the categorical/binary type of features and used scaled-PCA to generate better continuous features as appropriate input features for DNN. Thus, the proposed methodology can not only be applied to early detect OA with limited datasets but also be used for a diverse range of diseases with indirect statistical data.

For future work, we plan to modify and apply the proposed method on other medical utilization and health behavior datasets, such as those for stroke and dementia. In addition, we plan to use not only medical utilization and health behavior data but also detail decision parameters such as detailed index values or physiological signals, so that more continuous and meaningful information will be provided to the deep learning model as input data. Additionally, we will also apply auto-fine tuning methodologies to reduce the model training time and improve its overall performance.

## Figures and Tables

**Figure 1 ijerph-16-01281-f001:**
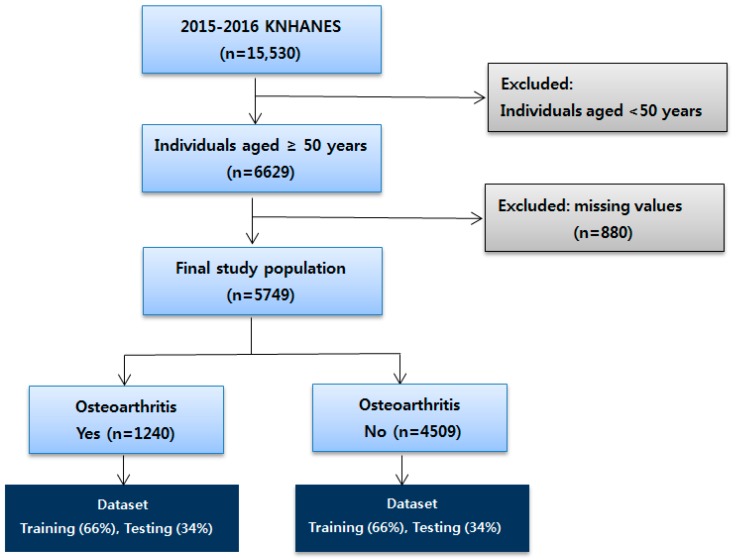
The selection process for the study population (KNHANES: Korea National Health and Nutrition Examination Survey).

**Figure 2 ijerph-16-01281-f002:**
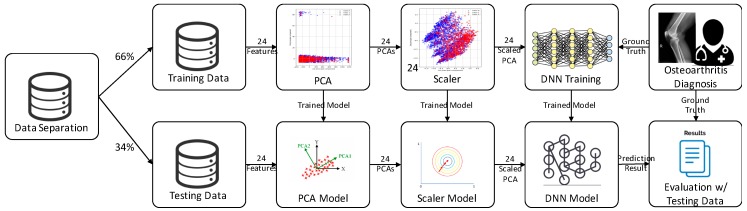
The proposed system architecture of the deep neural network (DNN) with scaled principal component analysis (PCA).

**Figure 3 ijerph-16-01281-f003:**
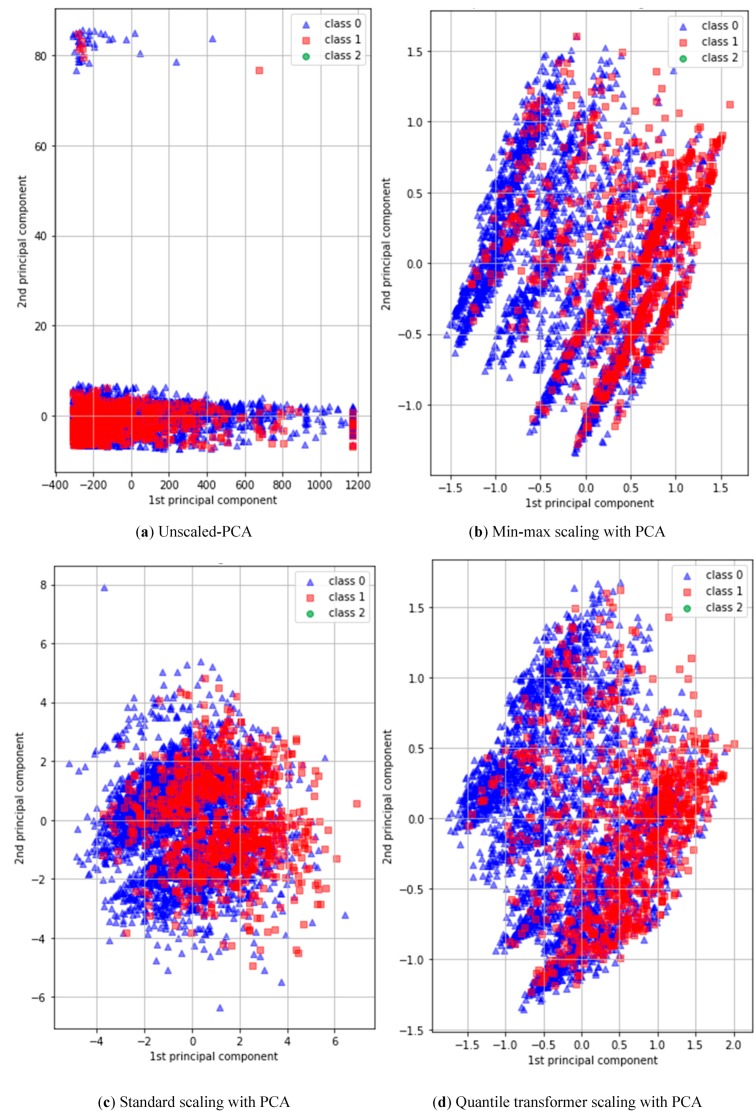
Two-dimensional plots of first and second principal components. (**a**) Unscaled-PCA. (**b**) Min-max scaling with PCA. (**c**) Standard scaling with PCA. (**d**) Quantile transformer scaling with PCA.

**Figure 4 ijerph-16-01281-f004:**
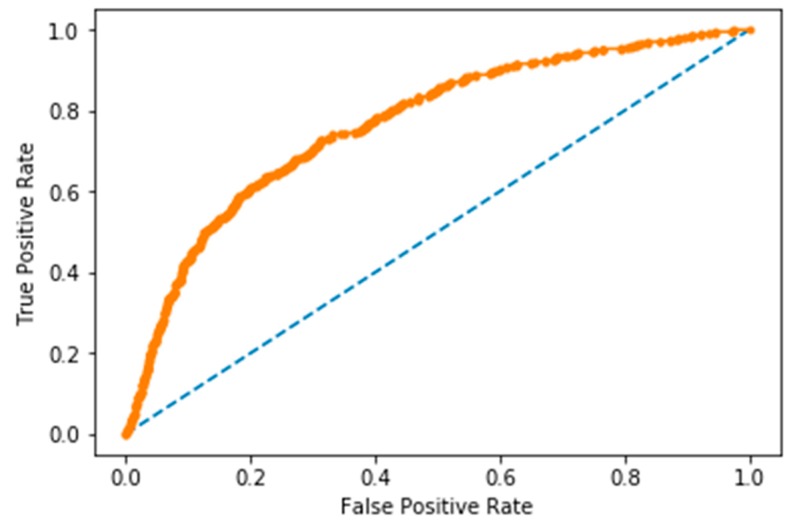
Receiver operating characteristic (ROC) curve for the predictive performance of DNN with scaled-PCA and area under the curve (*AUC*) of 76.8%.

**Table 1 ijerph-16-01281-t001:** Confusion matrix of the proposed method.

Confusion Matrix	Predicted (T)	Predicted (F)
Actual (T)	270	135
Actual (F)	413	1137

**Table 2 ijerph-16-01281-t002:** Correlation coefficient of initial features. BMI: body mass index.

Feature	Correlation Coefficient	Feature	Correlation Coefficient
Year	−0.023309	Angina	0.058126
Region	−0.017512	Osteoporosis	**0.254714**
Sex	**0.261386**	Diabetic mellitus	0.034071
Age	***0.208991***	Alcohol	-0.160021
Education	−0.24423	Smoking	0.190024
Household income	−0.158579	Physical activity	0.073899
Married	−0.008994	BMI	0.147746
Health status	***0.215611***	BMI group	0.14497
Hypertension	0.103292	Obesity	0.118041
Dyslipidemia	0.129874	Chronic disease count	***0.209151***
Stroke	0.029012	Region category	−0.012289
Myocardial infarction	0.016415	Income quartile	−0.182163

**Bold**: the top two best correlation values (±0.25), ***Italic***: the best correlation values (±0.2).

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
