# Peer review of "A Deep Neural Network-Based Method for Early Detection of Osteoarthritis Using Statistical Data"

_ijerph, 2019, doi:10.3390/ijerph16071281_

Round 1

Reviewer 1 Report

Title: A Deep Neural Network-Based Method for Early Detection of Osteoarthritis Using Statistical Data

This paper presents a deep neural network for detecting the occurrence of osteoarthritis using patient’s statistical data on medical utilization and health behaviour information. Moreover, the principal component analysis method with quantile transformer scaling was employed to generate the features from simple patient’s background medical records and identify the occurrence of osteoarthritis. In the stage of evaluation, the proposed method showed 76.8% of the area under the curve (AUC). Some sincerely comment as follows, for the author(s), may be useful to improve your work:

Abstract:

Comment 1: The authors should give readers some concrete information to get them excited about their work. The current abstract only describes the general purposes of the article. It should also include the article's main impact and significance of this manuscript for detection of osteoarthritis. 

Introduction: 

Comment 2: The authors should insert the statistic data of the number of case of the world population. 

Comment 3: I suggest to read and add in the introduction (as state of the art predictive of the model of the prevalence of osteoarthritis and machine learning) a clear discussion on the current literature (2015-2019) versus the unique contribution of the paper.

Comment 4: It is required to provide some including remarks to further discuss the proposed methods, for example, what are the main advantages and limitations in comparison with existing methods?

Methodology:

Comment 5: All parameters used should be indicated and discussed their tuning and effect on the results.

Comment 6: The proposed method tested on a dataset. What application results will be on the other datasets of the world data? More details should be furnished.

Results and Discussion:

Comment 5: The authors should provide the statistical significance test of the examples (features) used to represent each class of lesion. 

Comment 6: It is required to provide some concluding remarks to further discuss the proposed method for example, what are the main advantages and limitations in comparison with existing algorithms?

Conclusion:

Comment 7: In the conclusion section, the section is really much too short and the authors will have to demonstrate the impact and insights of the research. The authors need to rewrite the entire conclusion section with a focus on both impact and insights of the manuscript. No bullets should be used in your conclusion section.

Comment 8: If the paper is resubmitted as a significantly reworked piece of work, offering a proper view with clear Point-to-Point responses on what is the novelty and significantly improving the evaluation, then I can imagine a more positive second evaluation.

Author Response

We thank the editor and reviewers for their careful and thorough review. The comments and suggestions were very useful and helped to sharpen our thinking during the revision process. We have made following major changes to the manuscript based on the reviewer's suggestions.

In our original manuscript, we added related work section for comparing with previous literatures to check the novelty of our proposed method. We carefully read and reviewed fifteen papers regarding to the state-of-art research [16-30] on machine learning and deep learning methods between 2015 to 2019.

Following the suggestions by Reviewers 2 and 3, we added the overall system architecture of the proposed methodology with description and Figure 2 to make readers understand better.

We rewrote the conclusion and added several points about the advantage and limitation of this study in Introduction and discussion.

Point-by-point response to Reviewer 1's comments

Overall comment: This paper presents a deep neural network for detecting the occurrence of osteoarthritis using patient’s statistical data on medical utilization and health behaviour information. Moreover, the principal component analysis method with quantile transformer scaling was employed to generate the features from simple patient’s background medical records and identify the occurrence of osteoarthritis. In the stage of evaluation, the proposed method showed 76.8% of the area under the curve (AUC). Some sincerely comment as follows, for the author(s), may be useful to improve your work:

Response: We appreciate the reviewer’s comments and suggestions. Below please find our point-by-point response to your comments.

1.  [Abstract] The authors should give readers some concrete information to get them excited about their work. The current abstract only describes the general purposes of the article. It should also include the article's main impact and significance of this manuscript for detection of osteoarthritis.

Response: Thank you for pointing out this issue. We added several contents in the abstract Section as follow:

“A large number of people suffer from certain types of osteoarthritis, such as knee, hip, and spine. A correct prediction of osteoarthritis is an essential step to effectively diagnose and prevent severe osteoarthritis. Osteoarthritis is commonly diagnosed by experts through manually inspecting the patients’ medical images which are usually collected in hospitals. It is somewhat time-consuming for patients to simply check the occurrence of osteoarthritis. In addition, current studies focused on automatically detecting the osteoarthritis through image based deep learning algorithms. This needs patients’ medical images which requires patients to visit the hospital. However, medical utilization and health behavior information as a statistical data are easier to collect and access than medical images. It can be significant impacts on pro-active and preventive Medicare to predict diverse OAs occurrence using the indirect statistical data without any medical images. In this study, we used a deep neural network for detecting the occurrence of osteoarthritis using patient’s statistical data of medical utilization and health behavior information which is 5,749 of the total subject number. Principal component analysis with quantile transformer scaling are employed to generate the features from simple patient’s background medical records and identify the occurrence of osteoarthritis. Our experiments showed that the proposed method using deep neural network with the scaled-PCA showed 76.8% of area under curve (AUC) and minimize the effort to generate features. Hence, it can be a promising tool for patients and doctors to prescreen the possible osteoarthritis condition to reduce health cost and saving time to visit hospital.”

2.  [Introduction] The authors should insert the statistic data of the number of case of the world population.

Response: Thank you for pointing out this issue. We added the statistic data in the first paragraphs of Section 1 in the revised version.

The estimation of World Health Organization was that 18.0% of woman and 9.6% of men 60 years of age or over have symptomatic OA in 2015. Among people with symptomatic OA, 80% of them have some types of limitation in mobility and 25% are unable to perform their activities of daily living [3].”

[3] World Health Organization. 2016. [cited 2016 February 5].

Available from: http://www.int/chp/topics/rheumatic/en/.

3.  [Introduction] I suggest to read and add in the introduction (as state of the art predictive of the model of the prevalence of osteoarthritis and machine learning) a clear discussion on the current literature (2015-2019) versus the unique contribution of the paper.

Response: Thank you for pointing out this issue. We added the “2. Related Work” section based on current literature (2015-2019) about predictive models based on machine learning or deep learning for OA.

“2. Related Work

There have been several studies to use deep learning methodologies based on diverse data to detect several problems [16, 17, 18, 19, 20]. Especially, studies on detection or prediction of OA with machine learning or deep learning have been conducted with diverse approaches and datasets. Shaikhina et al. [21] proposed a NN model for osteoarthritic bone fracture risk stratification and decision tree model for prediction of antibody-mediated kidney transplant rejection. Although they achieved high accuracy, they used small datasets (35 bone specimens and 80 kidney transplants). Kovanova et al. [22] also investigated the trabecular bone in OA using NN with 25 available samples. Antony et al. [23] applied Convolution Neural Network (CNN) to quantify the severity of knee OA. They pre-trained dataset on ImageNet and turned parameters on knee OA images. Kobashi et al. [24] investigated a post-operative knee function prediction model using PCA based method with 2-D and 3-D X-ray data (52 OA patients). Ashinsky et al. [25] evaluated the ability of a machine learning algorithm to classify magnetic resonance images (MRI) of 65 OA patients. Lazzarini et al [26] proposed an analytics pipeline based on machine learning to predict the 30-months incidence of knee OA using diverse data including clinical variables, questionnaires, biochemical markers, and images. Xue et al [27] examined the diagnostic value of CNN with 420 hip X-ray images to detect hip OA. Tiulpin et al. [28] also used CNN to detect knee OA with X-ray imaged from 3,000 subjects. Du et al. [29] explored the features from knee MRI for OA prediction with PCA with several machine learning algorithms, such as NN, support vector machine, and random forest. Hirvasniemi applied machine learning to predict the incident radiographin hip OA using 986 images. Brahim et al. [30] applied machine learning (multivariate linear regression) to detect early knee IA using knee X-ray. Most of the previous studies focused on image data with machine learning or deep learning with image data, such as MRI or X-ray. However, to the best of our knowledge, the studies of prediction using statistical data with machine learning or deep learning are rare.”

4.  [Introduction] It is required to provide some including remarks to further discuss the proposed methods, for example, what are the main advantages and limitations in comparison with existing methods?

Response: Thank you for pointing out this issue. We have added the following paragraph in the Section 1.

“The present study is intended to develop a deep learning model to detect the occurrence of OA based on the statistical data of medical utilization and health behavior information. As advantages, most of the previous methods are relatively accurate and specific to the type of diseases. On the other hand, they have some limitations, such as inefficiency, uncomfortable, and time-consuming. More details about the previous work will be shown in Section 2.”

5.  [Methodology] All parameters used should be indicated and discussed their tuning and effect on the results.

Response: Thank you for pointing out this issue. We have added the following paragraph in the Section 4.

“We considered tuning the hyperparameters is the major process to implement the appropriate DNN for detecting OA, such as the number of node and depth of DNN. Although tuning the parameters is important, no general rules were determined so that we have to train many combinations, such as 4, 6, 8, and 10 layers with 70, 80, 90 and 100 nodes by trial and error. Another important issue is overfitting. We adopt two methods, such as Dropout and Batch normalization technique. Batch normalization is used to prevent vanishing feedforward data like a good weight initialization, and dropout is used to minimize the effects of certain hidden nodes with weights.”

6.  [Methodology] The proposed method tested on a dataset. What application results will be on the other datasets of the world data? More details should be furnished.

Response: Thank you for pointing out this issue. We have tried to find the other datasets of the world data, but there was no dataset that is similar one we used in this study. If the reviewer recommend the dataset, we can apply to our proposed methodology.

7.  [Results and Discussion] The authors should provide the statistical significance test of the examples (features) used to represent each class of lesion.

Response: Thank you for pointing out this issue. The methodology we used in this study is a deep neural network. In order to evaluate the developed model, we use five performance metrics, such as sensitivity, specificity, positive-predictivity, and accuracy. In addition, we used correlation coefficient test to check the statistical significance of each input feature. If the reviewer pointed out the p-value test with hypothesis, we didn’t conduct it because DNN is considered as a blackbox model and there was no results as p-value for the model. If there is another method that we can test it, please suggest the method. We will do that test.

8.  [Results and Discussion] It is required to provide some concluding remarks to further discuss the proposed method for example, what are the main advantages and limitations in comparison with existing algorithms?

Response: Thank you for pointing out this issue. We have added the following paragraph in the Section 4.

 “The proposed method predicts the occurrence of OA with indirect or limited data, such as the statistical data of medical utilization and health behavior information. This can be an advantage for possible patients to prevent future medical cost and reduce diagnosing time. There are some limitations such as, lack of the input data separation and no progressive data of OA. The study was based on a survey which had several defects. Most input features are binary type of data. Although we applied scaled-PCA to improve the data separation, additional feature inputs may be required. In addition, our results only apply to subjects with determined OA disease, since we excluded subjects who were receiving treatment for OA. This issue may reduce the overall accuracy of prediction model.”

9.  [Conclusion] In the conclusion section, the section is really much too short and the authors will have to demonstrate the impact and insights of the research. The authors need to rewrite the entire conclusion section with a focus on both impact and insights of the manuscript. No bullets should be used in your conclusion section.

Response: Thank you for pointing out this issue. We have tried to rewrite the conclusion section as follow:

“We proposed a deep learning model with scaled PCA for automatic osteoarthritis prediction based on medical utilization and health behavior data (5,749) without any hand-crafted features, and validated them in the large population. The most important finding of this study is the early detection of patients at high risk of OA who need additional checkup and appropriate treatment before exacerbation. The proposed method can also save a lot of time and efforts of designing, computing and selecting the features manually. Since the input data from the datasets is simple and low resolution like binary or limited number of categories, we employed DNN to learn the categorical/binary type of features and used scaled-PCA to generate better continuous type of features as appropriate input features for DNN. Thus, proposed methodology can be not only applied to early detect OA with limited datasets but also used to diverse disease with indirect statistical data.

For future work, we plan to modify and apply the proposed method on other medical utilization and health behavior data sets, such as stroke and dementia. In addition, we plan to use not only medical utilization and health behavior data but also detail decision parameters such as detailed index values or physiological signals, so that more continuous and meaningful information will be provided to the deep learning as input data. Additionally, we also will apply auto-fine tuning methodologies for reducing the model training time and improving overall performance.

10.  If the paper is resubmitted as a significantly reworked piece of work, offering a proper view with clear Point-to-Point responses on what is the novelty and significantly improving the evaluation, then I can imagine a more positive second evaluation.

Response: Thank you for pointing out many valuable issues. We have tried to add all point-to-point responses. If you need any other issues, we can happily work on them.

Reviewer 2 Report

This article proposes early detection of osteoarthritis, which is based on DNN. Statistical data from KCDCP as their training data.

This article has a good introduction to explain the background. However, this article lacks the system block diagram (system architecture) for their proposed method. 

Figure 1 should be redrawn. Many lines have not connected with their mapped blocks.

For equations, the definition of variables (FN, TP, TN) should be described firstly. Moreover, these math variables should be "italic"

This article lacks a comparison with related works. There are many DNN-based methods have been published. Why authors adopted this DNN architecture. Detailed analysis for selecting this DNN architecture should also be discussed. 

Author Response

We thank the editor and reviewers for their careful and thorough review. The comments and suggestions were very useful and helped to sharpen our thinking during the revision process. We have made following major changes to the manuscript based on the reviewer's suggestions.

In our original manuscript, we added related work section for comparing with previous literatures to check the novelty of our proposed method. We carefully read and reviewed fifteen papers regarding to the state-of-art research [16-30] on machine learning and deep learning methods between 2015 to 2019.

Following the suggestions by Reviewers 2 and 3, we added the overall system architecture of the proposed methodology with description and Figure 2 to make readers understand better.

We rewrote the conclusion and added several points about the advantage and limitation of this study in Introduction and discussion.

Other changes will be elaborated in our responses to reviewers’ comments.

Point-by-point response to Reviewer 2's comments

Overall comment: This article proposes early detection of osteoarthritis, which is based on DNN. Statistical data from KCDCP as their training data:

Response: We appreciate the reviewer’s comments and suggestions. Below please find our point-by-point response to your comments.

1.  This article has a good introduction to explain the background. However, this article lacks the system block diagram (system architecture) for their proposed method.

Response: Thank you for pointing out this issue. We added the system architecture diagram in Figure 2 and the description in the second paragraph of Section 3.3 in the revised version.

Figure 2 shows the proposed system architecture of the DNN with scaled-PCA: (1) KNHANES data, including 24 raw features divided into training (66%) and testing data (36%); (2) data preprocessing based on the PCA and scaler in applied to convert categorical variables into continuous variables and generates models for testing data; (3) DNN model is also trained based on scaled PCA variables and generates the trained DNN model; (4) The predicted results are evaluated with the ground truth data labeled by clinicians. We have divided the training and testing data which means there is no overlapped patient data.

Figure2. The proposed system architecture of the DNN with scaled-PCA.

2.  Figure 1 should be redrawn. Many lines have not connected with their mapped blocks.

Response: Thank you for pointing out this issue. We redraw the Figure 1.

Figure1. The selection process for the study population (KNHANES: Korea National Health and Nutrition Examination Survey).

3.  For equations, the definition of variables (FN, TP, TN) should be described firstly. Moreover, these math variables should be "italic"

Response: Thank you for pointing out this issue. We added the description in the thrid paragraph of Section 3.3 in the revised version.

There are four terms, such as true positives (TP), true negatives (TN), false positives (FP) and false negatives (FN), to compute the performance parameters to evaluate the developed models. TP is the correctly predicted OA and TN is the correctly predicted non-OA. FP is the incorrectly predicted OA and FN is the incorrectly predicted non-OA.”

4.  This article lacks a comparison with related works. There are many DNN-based methods have been published. Why authors adopted this DNN architecture. Detailed analysis for selecting this DNN architecture should also be discussed.

Response: Thank you for pointing out this issue. We added the “2. Related Work” section based on current literature (2015-2019) about predictive models based on machine learning or deep learning for OA.

2. Related Work

There have been several studies to use deep learning methodologies based on diverse data to detect several problems [16, 17, 18, 19, 20]. Especially, studies on detection or prediction of OA with machine learning or deep learning have been conducted with diverse approaches and datasets. Shaikhina et al. [21] proposed a NN model for osteoarthritic bone fracture risk stratification and decision tree model for prediction of antibody-mediated kidney transplant rejection. Although they achieved high accuracy, they used small datasets (35 bone specimens and 80 kidney transplants). Kovanova et al. [22] also investigated the trabecular bone in OA using NN with 25 available samples. Antony et al. [23] applied Convolution Neural Network (CNN) to quantify the severity of knee OA. They pre-trained dataset on ImageNet and turned parameters on knee OA images. Kobashi et al. [24] investigated a post-operative knee function prediction model using PCA based method with 2-D and 3-D X-ray data (52 OA patients). Ashinsky et al. [25] evaluated the ability of a machine learning algorithm to classify magnetic resonance images (MRI) of 65 OA patients. Lazzarini et al [26] proposed an analytics pipeline based on machine learning to predict the 30-months incidence of knee OA using diverse data including clinical variables, questionnaires, biochemical markers, and images. Xue et al [27] examined the diagnostic value of CNN with 420 hip X-ray images to detect hip OA. Tiulpin et al. [28] also used CNN to detect knee OA with X-ray imaged from 3,000 subjects. Du et al. [29] explored the features from knee MRI for OA prediction with PCA with several machine learning algorithms, such as NN, support vector machine, and random forest. Hirvasniemi applied machine learning to predict the incident radiographin hip OA using 986 images. Brahim et al. [30] applied machine learning (multivariate linear regression) to detect early knee IA using knee X-ray. Most of the previous studies focused on image data with machine learning or deep learnig. However, to the best of our knowledge, the studies of prediction using statistical data with machine learning or deep learning are rare.”

Reviewer 3 Report

This paper proposed a DNN-based approach for early osteoarthritis detection, which is adopted related patient’s statistical data of medical utilization and health behavior information. First of all, this paper tried to solve an important issue, which is very interesting. Moreover, the authors gave a good introduction to the background of this issue.

However, the main problem of this paper is that the whole system architecture is not clear, hence, I recommend that the authors should explain their system architecture and related implemented methods.

Figure 1 should be improved, some lines have not connected between some blocks.

The content of this paper is not enough, the authors should be introduced and discussed their work in details. Please note that this journal has not pages limitation.

Author Response

We thank the editor and reviewers for their careful and thorough review. The comments and suggestions were very useful and helped to sharpen our thinking during the revision process. We have made following major changes to the manuscript based on the reviewer's suggestions.

In our original manuscript, we added related work section for comparing with previous literatures to check the novelty of our proposed method. We carefully read and reviewed fifteen papers regarding to the state-of-art research [16-30] on machine learning and deep learning methods between 2015 to 2019.

Following the suggestions by Reviewers 2 and 3, we added the overall system architecture of the proposed methodology with description and Figure 2 to make readers understand better.

We rewrote the conclusion and added several points about the advantage and limitation of this study in Introduction and discussion.

Point-by-point response to Reviewer 3's comments

Overall comment: This paper proposed a DNN-based approach for early osteoarthritis detection, which is adopted related patient’s statistical data of medical utilization and health behavior information. First of all, this paper tried to solve an important issue, which is very interesting. Moreover, the authors gave a good introduction to the background of this issue.

Response: We sincerely thank you for your time and efforts in reading and commenting our paper. Below please find our responses in detail.

1. However, the main problem of this paper is that the whole system architecture is not clear, hence, I recommend that the authors should explain their system architecture and related implemented methods.

Response: Thank you for pointing out this issue. We added the system architecture diagram in Figure 2 and the description in the second paragraph of Section 3.3 in the revised version.

Figure 2 shows the proposed system architecture of the DNN with scaled-PCA: (1) KNHANES data, including 24 raw features divided into training (66%) and testing data (36%); (2) data preprocessing based on the PCA and scaler in applied to convert categorical variables into continuous variables and generates models for testing data; (3) DNN model is also trained based on scaled PCA variables and generates the trained DNN model; (4) The predicted results are evaluated with the ground truth data labeled by clinicians. We have divided the training and testing data which means there is no overlapped patient data.

Figure2. The proposed system architecture of the DNN with scaled-PCA.”

2. Figure 1 should be improved, some lines have not connected between some blocks.

Response: Thank you for pointing out this issue. We redraw the Figure 1.

Figure1. The selection process for the study population (KNHANES: Korea National Health and Nutrition Examination Survey).

3. The content of this paper is not enough, the authors should be introduced and discussed their work in details. Please note that this journal has not pages limitation.

Response: Thank you for pointing this out. We have added many sentences and paragraphs in several Sections, such as XXXX.

Round 2

Reviewer 1 Report

Title: A Deep Neural Network-Based Method for Early Detection of Osteoarthritis Using Statistical Data

The work presents a deep neural network-based method for early detection of osteoarthritis using statistical data. The authors have addressed the remarks in my suggestions of review. This paper has improved based in the suggestions presented in the manuscript. I recommend acceptance of the work in the journal.

Reviewer 2 Report

The authors have great efforts for enhancing and improving their first submitted draft.

The authors have satisfyingly replied to my all questions. So, I think that the revised manuscript can be accepted for publication.

Reviewer 3 Report

The revised paper can be accepted for publication.